# Clinical, Bone Mineral Density and Spinal Remodelling Responses to Zoledronate Treatment in Chronic Recurrent Multifocal Osteomyelitis

**DOI:** 10.3390/diagnostics15182320

**Published:** 2025-09-12

**Authors:** Fahim Patel, Penelope J.C. Davis, Nicola Crabtree, Suma Uday

**Affiliations:** 1Department of Paediatric Rheumatology, Birmingham Women’s and Children’s Hospital, Birmingham B4 6NH, UK; fahim.patel@nhs.net (F.P.);; 2Department of Endocrinology, Birmingham Women’s and Children’s Hospital, Birmingham B4 6NH, UK; 3Department of Metabolism and Systems Science, University of Birmingham, Birmingham B15 2TT, UK

**Keywords:** bisphosphonates, CRMO, CNO, SAPHO, spine, vertebral, osteomyelitis

## Abstract

**Background:** Chronic Recurrent Multifocal Osteomyelitis (CRMO) is a rare auto-inflammatory condition affecting the growing skeleton. The standard first-line treatment of high-dose NSAIDs (non-steroidal anti-inflammatory drugs) is adequate only in a subset of patients. The American College of Rheumatology Consensus Guidelines suggest considering bisphosphonates in a certain category of patients based on evidence from a handful of case series reporting the outcome of pamidronate use. **Aims:** The aim of this study was to report the efficacy and safety of bisphosphonate, predominantly zoledronate, use in CRMO. **Methods:** A retrospective cohort study of children with CRMO receiving bisphosphonates was conducted between January 2008 and September 2023 at a single tertiary referral centre. We described the baseline characteristics; clinical indication, regimen and response to bisphosphonate treatment; changes in bone mineral density (BMD) and spine remodelling on dual-energy X-ray absorptiometry (DXA) scans; and safety data. **Results:** During the study period, 64 (72%, *n* = 46 females) patients with CRMO with a median age at diagnosis of 10 years (range: 3 to 16 years) were identified. Approximately 31% (*n* = 20) received either pamidronate (*n* = 2) or zoledronate (*n* = 14) or both (*n* = 4) due to changes in local protocols. The most frequent indications for bisphosphonate use were refractory pain [55%, *n* = 11/20], pain + spine involvement [35% (*n* = 7/20)] and spine involvement only [10% (*n* = 2)]. Prior to bisphosphonate therapy, 100% took regular NSAIDs (*n* = 19/19), 21% (*n* = 4/19) used opioids, 47% (*n* = 9/19) received oral steroid courses, and 10% (*n* = 2/19) received methotrexate. The median age at bisphosphonate treatment initiation was 12 years (range 6–18 years), and the duration of treatment was 2 years (range: 6 months to 5 years). Improvement in pain was reported by 88% of patients (*n*= 15/17, 1 was excluded as they had not started treatment yet). All non-responders (*n* = 2/17;) to bisphosphonate therapy were later recognised clinically to have pain amplification syndrome and were referred to the chronic pain multi-disciplinary team. This correlated to the complete treatment de-escalation of opioids (*n* = 3/3; 1 was excluded as they had not yet started treatment), steroids (*n* = 8/8) and methotrexate (*n* = 2/2). NSAIDs were discontinued in 44% of patients (*n* = 7 of 16; 1 was excluded due to missing data, and 3 were excluded due to NSAID intolerance). The median first-year increase in the LS BMAD (lumbar spine bone mineral apparent density) Z-score was +1.35, and that in the TBLH BMD (total body less head bone mineral density) Z-score was +0.7 (*n* = 11). Subsequently, median average annual increases in the LS BMAD Z-score of +0.65 and in the TBLH BMD Z-score of +0.45 (*n* = 5) were recorded. Around 30% of patients (*n* = 6) required treatment modification (dose reduction, frequency reduction or cessation) due to a rapid escalation in BMD. There were no fractures documented due to raised BMD. Evidence of spine remodelling on DXA vertebral fracture assessment was seen in 38% of patients with spinal lesions (*n* = 3 of 8). There was no radiological evidence of improvement in any vertebra plana lesion. First-phase reactions (pyrexia) were reported universally in patients who received bisphosphonates, but none were significant requiring hospitalisation. **Conclusions:** Similar to pamidronate, zoledronate with an advantageous dosing regimen is well tolerated and effective in improving pain and enabling the de-escalation of adjunctive therapy in CRMO. This is the first report tracking changes in BMD and spinal remodelling in response to zoledronate in CRMO patients. Spinal remodelling is minimal in vertebra plana lesions. Bone density monitoring and personalisation of the bisphosphonate dose and regimen are strongly recommended to avoid overtreatment.

## 1. Introduction

Chronic Recurrent Multifocal Osteomyelitis (CRMO) [or Chronic Non-Bacterial Osteomyelitis (CNO) if referring to the same disease with a single lesion] is a rare skeletal auto-inflammatory condition affecting the growing skeleton in children and adolescents. It was first described in 1972 as a subacute and chronic “symmetrical” osteomyelitis of unknown aetiology [1]. The exact demographics are unknown; incidence is estimated at 0.65 per 100,000, with a prevalence of 1–9 per 100,000. The reported incidence is likely greatly underestimated owing to challenges in the recognition and diagnosis of the condition [2].

The majority of patients have an innocuous presentation mimicking growing pains commonly seen in this age group. The “classic” presentation of CRMO is a unilateral painful mass of the middle-third of the clavicle, seen in only 35% of patients in a recent BPSU (British Paediatric Surveillance Unit) study [3]. The most common sites however are lower limb long bones. Other disease sites include long bones of the upper limbs, spine, pelvis and jaw [3]. Interestingly a CRMO-like presentation is also seen in SAPHO (synovitis, acne, pustulosis, hyperostosis and osteitis) and in the monogenic mimics Majeed Syndrome (LPIN2 mutation) and Deficiency of Interleukin-1 Receptor Antagonist Syndrome (IL1RN mutation).

The first investigation in the diagnosis of CRMO is usually a plain-film X-ray, which may show bony sclerosis, lytic lesions and metaphyseal lesions in the long bones. The main differentials for CRMO include malignancy, Langerhans’ Cell Histiocytosis (if the spine/skull is involved) or infection. Advanced diagnostic techniques such as magnetic resonance imaging (regional and/or whole body), bone scintigraphy and tissue sampling are usually required to reach a diagnosis. In a UK-wide surveillance study published in 2024, 51.1% of patients with CRMO underwent a bone biopsy to establish their diagnosis [3].

In the absence of a diagnosis and/or treatment, the natural progression is usually the self-resolution of inflammation on reaching skeletal maturity [4]. During this period, patients are prone to secondary complications, such as pain, limb fractures, vertebral lesions, kyphosis, growth arrest and/or leg length discrepancy [5]. The main treatment goals in CRMO are pain management and the prevention of secondary complications. The standard first-line treatment is high-dose NSAIDs (non-steroidal anti-inflammatory drugs), which is effective in a subset of patients. However, second-line treatment options, where pain is uncontrolled, are less clear. Furthermore, the pathogenesis of CRMO is not well understood, complicating the development of treatment strategies.

Our current knowledge postulates an autoinflammatory process in part driven by Interleukins (ILs), particularly IL-1 and IL-10. This view is supplemented by the emergence of monogenic mimics, which exhibit common dysfunction in IL-1 antagonism; this is a key driver in the inflammatory cascade resulting in skin and bone inflammation with a presentation similar to CRMO. Bisphosphonates were first suggested as a treatment option because osteoclasts are considered effector cells of this inflammatory cascade.

Consensus Treatment Guidelines published by the ACR (American College of Rheumatology) in 2019 suggest that patients with pain refractory to NSAIDs or those with spinal lesions should be considered for bisphosphonate therapy [6]. This level 4 evidence for bisphosphonate use in CRMO comes from seven retrospective reports of pamidronate use, in varying regimens and duration, in cohorts containing <11 patients [7,8,9,10,11,12,13]. There is little data on the effect of bisphosphonate use in CRMO on bone mineral density (BMD), a condition where the baseline BMD is usually normal. Prolonged bisphosphonate use is associated with an iatrogenic osteopetrosis-like state [14], which necessitates monitoring with interval bone density scans to avoid overtreatment. The optimum bisphosphonate regimen that addresses pain and potentially aids in the healing of CRMO lesions whilst avoiding an excessive rise in bone mineral density is not known.

## 2. Aims

The aim of this study was to evaluate the response to treatment with bisphosphonates, predominantly zoledronate, in a cohort of patients with CRMO.

## 3. Objective

The objective of this study was to evaluate the response to bisphosphonate treatment in terms of pain, changes in BMD and spinal remodelling over the treatment course.

## 4. Methods

This was a retrospective cohort study of paediatric patients with CRMO from a single tertiary centre, of whom a subset was treated with bisphosphonates between 2008 and 2023. The response to treatment with regards to pain, spinal remodelling and BMD is reported.

### 4.1. Study Setting

The specialist clinic at Birmingham Women’s and Children’s Hospital (BWCH) is a tertiary referral centre for paediatric rheumatology and endocrinology. Referrals to the clinic are from paediatric rheumatologists, who, in turn, receive referrals from general paediatricians, trauma and orthopaedics and bone oncology multi-disciplinary teams. Referrals to the clinic are made for the consideration of bisphosphonate therapy in the event of failure of conventional CRMO treatment, uncontrolled pain and/or the presence of spinal lesions.

### 4.2. Treatment Protocol

Patients were treated as per local bisphosphonate protocols. Until 2018, patients received pamidronate at a total dose of 9 mg/kg/year administered every 4–6 months over 2 consecutive days each cycle. Patients initiating bisphosphonates after 2014 received zoledronate due to changes in local protocols. Zoledronate was administered every 6 months at an annual dose of 0.1 mg/kg. Our bisphosphonate dosing was in keeping with the consensus guidelines on the use of bisphosphonate therapy in children and adolescents [15]. All patients received supplemental calcium (50 mg/kg/day, max 1 g/day) for 7 days following treatment as per local protocols.

### 4.3. Subjects

This was a retrospective study of all patients with CRMO referred to a specialist joint paediatric and rheumatology clinic at Birmingham Women’s and Children’s Hospital between January 2008 and December 2023.

### 4.4. Data Gathering

Patients with CRMO were identified retrospectively from electronic case records using diagnosis coding data from DAWN (4S-DAWN Rh Rheumatology software). We reviewed electronic patient records on PEPR (Paediatric Electronic Patient Record software) to identify a subset of CRMO patients who were referred for bisphosphonate therapy. We recorded baseline demographics, lesions at diagnosis, treatments to date and the clinical indication for referral for bisphosphonate therapy. Information was gathered on response to pain. Pre-treatment (baseline) and annual DXA (dual-energy X-ray absorptiometry) scan results were gathered for the quantification of the LS BMAD (lumbar spine bone mineral apparent density), TBLH BMD (total body less head bone mineral density) and DXA VFA (vertebral fracture assessment) for spinal abnormalities.

### 4.5. Statistics

Descriptive statistics are used to report patient demographics, responses to pain and changes in BMD from baseline.

### 4.6. Ethics

This retrospective audit was registered locally using the Clinical Audits and Registry Management Service (CARMS); the assigned identification number is 31920.

## 5. Results

### 5.1. Baseline Characteristics

Between January 2008 and September 2023, 64 patients with CRMO were identified from electronic records; 72% (*n* = 46) were female. The median age at diagnosis was 10 years (range: 3–16), with a median of 4 lesions per patient (range: 1–17). The anatomical locations of the lesions were as follows: 42% were in the lower limbs, 28% were in the spine, 10% were in the pelvis, 9% were in the upper limbs, 8% were in the clavicle, 2% were in the sternal/chest wall, and 1% were in the jaw. The baseline characteristics and treatment of CRMO patients referred for bisphosphonate therapy are summarised in detail in Table 1.

### 5.2. Indication for Treatment

Of the 64 patients, approximately 31% (*n* = 20) received bisphosphonate therapy. The majority received bisphosphonates for refractory pain (without spinal involvement) [55%, *n* = 11/20], 35% (*n* = 7/20) received them for pain and spinal involvement, and 10% (*n* = 2) received them for spine involvement only (Table 1).

### 5.3. Pre-Bisphosphonate Treatment

Prior to bisphosphonate therapy, 100% took regular NSAIDs (*n* = 19/19; 1 was excluded due to missing data), 21% (*n* = 4/19) used opioids, 47% (*n* = 9/19) received intermittent courses of oral steroids, and 10% (*n* = 2/19) were on methotrexate (Table 1).

### 5.4. Bisphosphonate Treatment Details

The median age at the start of treatment was 12 years (range 6–18 years). The median treatment duration was 2 years (range: 6 months to 5 years). Two patients (10%) received pamidronate alone, fourteen (70%) received zoledronate alone, and four (20%) switched from pamidronate to zoledronate during the period of changes in local protocols, thus receiving both (Table 1).

### 5.5. Treatment Response

The clinical response to treatment (pain and the de-escalation of analgesics), spinal remodelling and changes in BMD on DXA scans in individual patients are summarised in Table 1.

### 5.6. Clinical Response

Improvement in pain was reported by 88% of patients (*n*= 15/17, 1 was excluded as they had not started treatment yet). This corresponded to the complete treatment de-escalation of opioids (*n* = 3/3; 1 was excluded as they had not yet started treatment), steroids (*n* = 8/8; 1 was excluded due to pain amplification syndrome) and methotrexate (*n* = 2/2). NSAIDs were discontinued in 44% of patients (*n* = 7 of 16; 1 was excluded due to missing data, and 3 were excluded due to NSAID intolerance).

All non-responders (*n* = 2/17;) to bisphosphonate therapy were later recognised clinically to have pain amplification syndrome and were referred to the chronic pain multi-disciplinary team.

Three patients required treatment escalation during or after starting bisphosphonate therapy. Patient 1 had a rapidly escalating BMD in combination with NSAID intolerance; for renal concerns, they were commenced on methotrexate and, in retrospect, were suspected to have a monogenic mimic of CRMO. Patient 3 was prescribed an anti-TNF biologic and was also suspected to have a monogenic mimic of CRMO. Patient 20 was commenced on methotrexate 2 years after stopping treatment with bisphosphonates.

### 5.7. Change in BMD

In the cohort where baseline and one-year follow-up scans were available (*n* = 11), there was a median first-year increase in the LS BMAD (lumbar spine bone mineral apparent density) Z-score of +1.35 and in the TBLH BMD (total body less head bone mineral density) Z-scores of +0.7. Beyond the first year of treatment, there was a median average annual increase in the LS BMAD Z-score of +0.65 and in the TBLH BMD Z-scores of +0.45 (*n* = 5).

### 5.8. Radiographic Remodelling Response

The spinal remodelling response to bisphosphonate treatment for each individual patient is summarised in Table 1; examples of response/non-response are illustrated in Figure 1. Radiological evidence of spinal remodelling was seen in 38% of patients with spinal lesions (*n* = 3 of 8). There was no radiological evidence of improvement in any vertebra plana lesion.

### 5.9. Safety

A rapid escalation in BMD requiring treatment modification (either a dose reduction, frequency reduction or cessation) was seen in 30% of patients (*n* = 6), of whom 50% (*n* = 3/6) had their treatment ceased. Individual patient responses and actions are summarised in Table 1. There were no fractures documented with rapid rises in BMD or supraphysiological BMD.

First-phase reactions (pyrexia) were reported universally in patients who received bisphosphonates, but none were significant requiring hospitalisation.

One patient had an extravasation injury without any additional treatment required.

## 6. Discussion

To the best of our knowledge, this is the first report of the response to zoledronate treatment in a large cohort of children and adolescents with CRMO and the first to document the serial change in BMD and spinal remodelling response. Zoledronate, with a convenient dosing regimen compared to pamidronate, is effective in managing pain and de-escalating adjunctive treatment in CRMO. Spinal remodelling is noted with a mild-to-moderate degree of vertebral deformities but not with vertebra plana. Given the escalation in bone density noted with zoledronate treatment in CRMO, a starting dose lower than that used in osteoporotic conditions is warranted, along with annual monitoring of BMD and personalisation of the treatment dose/regimen.

The baseline characteristics of our cohort were similar to the national CRMO cohort reported in the recent British Paediatric Surveillance Unit (BPSU) survey [3]. Our findings of a median age at diagnosis of 10 years and predominance in females (72%) are in keeping with the BPSU data. However, there was a higher incidence of patients with spinal lesions (28% vs. 18.5%) and a higher median number of lesions (4 vs. 2) in our cohort, which may reflect the nature of referrals to our tertiary unit. Moreover, patients managed in peripheral units requiring escalation to bisphosphonate therapy are often referred to our centre for treatment under expert guidance.

The main indication for escalation to bisphosphonate therapy in our cohort was pain, with or without spinal involvement, with a minority being referred for spinal involvement alone. These indications are in line with the consensus treatment guidelines, but there are no direct comparator studies reporting on the proportion of patients who receive bisphosphonate therapy and the clinical indication.

Prior to bisphosphonates, we found that all who were tolerant of NSAIDs were treated with such. In our cohort, 47% received intermittent steroids, 21% received opioids, and 10% received DMARDs. The recent BPSU survey, which reported only on the initial management of CRMO, found that 93.9% were prescribed NSAIDs, 44.8% were prescribed bisphosphonates, 6.5% were prescribed opioids, 5.9% were prescribed steroids, and 3% were prescribed methotrexate [3]. The study did not report on the use of combination therapies of the above [3]. Of note, 44.8% of patients were reported to have received bisphosphonates as their initial management, of whom only 8% received zoledronate, and the rest received pamidronate. The survey did not detail the indication for bisphosphonates, the treatment regimen adopted or the outcome [3].

The bisphosphonate treatment regime used for our patients was either pamidronate at an annual dose of 9 mg/kg administered every 4–6 months over 2 consecutive days each cycle or zoledronate administered every 6 months at an annual dose of 0.1 mg/kg, a regimen frequently used in children and adolescents with osteoporosis and other metabolic bone conditions where bisphosphonate treatment is indicated [15]. The rheumatology consensus guidelines offer recommendations for pamidronate as either monthly or quarterly infusions with a maximum annual cumulative dose of 11.5 mg/kg based on evidence from the seven case series included in the consensus [6]. Although no studies had reported on the use of zoledronate at the time of the consensus reporting, a quarterly or biannual dosing with a maximum of 0.1 mg/kg/year and a lower initial dosing of 0.0125–0.025 mg/kg was recommended [15]. The first use of zoledronate (0.025 mg/kg every 3 months for 12 months) in CRMO was reported in a conference abstract [16] to treat spinal lesions in three children. More recently, an adult study utilising zoledronate infusion in 24 patients with CNO of the jaw reported using on average 4.1 infusions (median of 3) per year of 4 mg each for an average treatment period of 33.4 months (median of 23) [17]. These studies did not monitor or report bone density changes in response to treatment.

The clinical response to bisphosphonates in our cohort was assessed by the de-escalation of adjunctive treatment and improvement in reported pain. The impact of bisphosphonates on pain was remarkable in all but two responders, who were later diagnosed with pain amplification syndrome. Bisphosphonates facilitated the treatment de-escalation of adjunctive therapy, including methotrexate, steroids and opioids. The majority of the cohort did however continue to need NSAIDs to some degree.

The changes in BMD during the treatment course were striking; the greatest rise was in the spine density Z-score in the first year of treatment, with an increase of +1.35. Whole-body density also showed a modest rise of +0.7. Beyond the first year of treatment, the increase was modest, with rises in the lumbar spine density Z-scores of +0.65 and +0.45 for the whole body. Six patients demonstrated a rapidly escalating BMD, of whom five reached supraphysiological states. The only comparator study to this is Gleeson et al.’s [10] cohort, who, after treatment with pamidronate, demonstrated a BMD increase in L2–L4 from –1.9 (1.0) to +0.9 (0.4) and in TBLH from –1.2 (1.5) to +0.1 (0.9). This was however limited to seven patients, with a median treatment duration of 12 m (6–41 m). The consensus guidelines do not make any specific recommendations for DXA scanning but advise on gathering this information where available [6].

The spinal remodelling response remains poorly researched at present. A couple of reports on the response to pamidronate are noted in the literature. Gleeson et al. [11] showed evidence of vertebral remodelling with pamidronate in three of seven cases reported. Hospach et al. [12] reported an improvement in vertebral height in three vertebrae in two patients treated with pamidronate. Robinson et al. [16] reported the resolution of hyperintense spinal lesions seen on MRI following treatment in two patients; however, a child with a grade 3 vertebral fracture showed no improvement at six months. The effect of bisphosphonates on spinal remodelling in our cohort was similar, with no demonstrable impact on vertebra plana and only mild-to-moderate deformities showing improvement.

There are safety concerns when bisphosphonates are used in patients with a normal BMD, increasing the risk of inducing an iatrogenic osteopetrosis-like state [14]. Importantly, none of the patients in this sub-group were found to have complications from overtreatment such as fractures. Nonetheless, a cautious approach to dosing is recommended, alongside regular bone density monitoring. A reduction in the annual dose or frequency is recommended when an escalation in bone density is recorded.

First-phase reactions were universal in those receiving bisphosphonates; this was well tolerated and did not require treatment modification. This is a well-documented phenomenon, which did not require treatment modification.

### Limitations and Future Work

Given the retrospective nature of this study, there are several limitations; nonetheless, this is the largest cohort of children and adolescents with CRMO treated with zoledronate. The assessment of pain pre- and post-infusion consisted of a descriptive report by the patients, as documented in the notes, and a validated pain score before and after treatment was not employed. Furthermore, some of the vertebral remodelling analyses used different imaging modalities, for example, pre-treatment MRI vs. follow-up VFA. A uniform imaging modality for pre- and post-treatment assessments would be beneficial for systematic comparisons. Not all patients had baseline and annual follow-up bone density scans; nonetheless, the need for such scans was established based on the response to treatment seen in our cohort. Future studies should aim to establish the optimum dose of bisphosphonates required for pain and lesion management without causing any unnecessary escalation in BMD. There is a paucity of data on the long-term effects of bisphosphonates in this cohort; this is particularly important given the very long half-life of bisphosphonates [18].

## 7. Conclusions

This is the first report of a large cohort of patients with CRMO managed predominantly with zoledronate, where the effects on BMD and spinal remodelling were longitudinally tracked. Similar to pamidronate, zoledronate is well tolerated and effective in CRMO patients with an advantageous dosing regimen. Bisphosphonates improved pain and enabled the de-escalation of adjunctive therapy, including DMARDs (disease-modifying anti-rheumatic drugs), NSAIDs, opioids and steroids. Spinal remodelling was minimal, with no improvement in vertebra plana. We recommend using doses of bisphosphonates lower than those used in osteoporotic conditions and personalising treatment based on baseline and serial bone densitometry, alongside assessing the clinical response to pain and lesion progression.

## Figures and Tables

**Figure 1 diagnostics-15-02320-f001:**
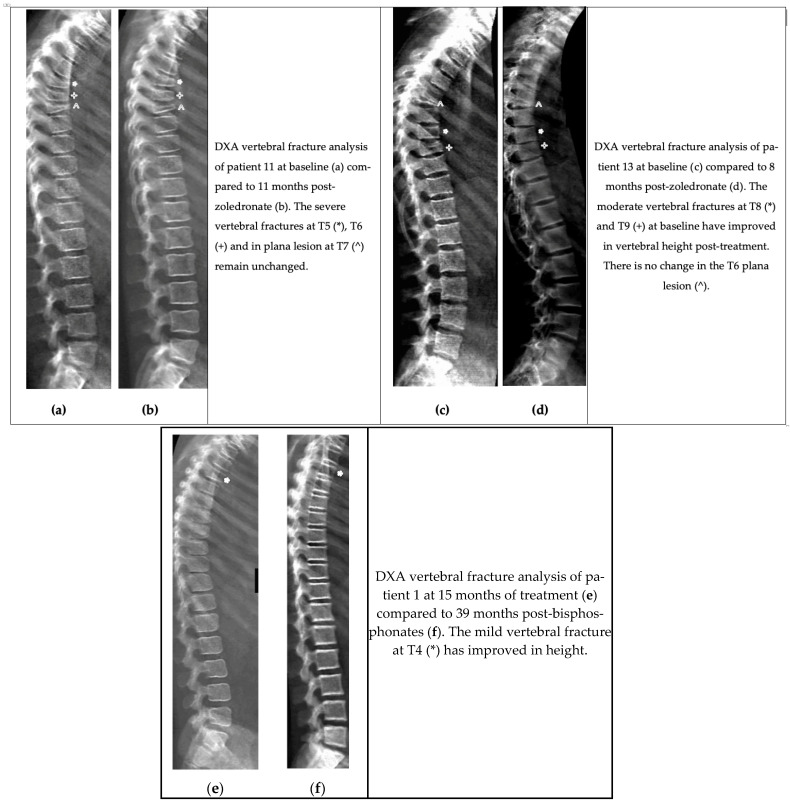
Examples of spinal response to bisphosphonates in CRMO-associated vertebral fractures.

**Table 1 diagnostics-15-02320-t001:** Baseline characteristics and treatment summary of CRMO patients referred for bisphosphonate therapy.

Patient Details		Lesions	Pre-BP Treatment	BP Indication	Treatment Details	Clinical Response	Spinal Response	Year 1 Delta BMD	Post-Year 1 Annual Delta BMD Average
1	9y Male	Mild spinal 1 Upper limb 3 Lower limb 3 Pelvic 5	NSAID Steroids	Pain	4 years of treatment with pamidronate then zoledronate Treatment was stopped due to escalating BMD Suspected monogenic mimic of CRMO	Improved pain Treatment de-escalation	Yes	No baseline	LS BMAD +0.45 per year TBLH BMD +0.25 per year
2	14y Female	Mild spinal 1 Lower limb 1 Pelvic 2	NSAID Steroids	Pain	1 year of Treatment with Zoledronate Stopped due to Inefficacy Repeat MRI showed no active CRMO lesions Suspected pain amplification syndrome	None	N/A (sacral)	LS BMAD +0.1 TBLH BMD + 0.5	N/A
3	15y Female	Mild spinal 5 Upper limb 2 Lower limb 5 Pelvic 2	NSAID Steroids	Spinal Lesions Pain	1 year of Treatment with Zoledronate Stopped BP for inefficacy Suspected monogenic mimic Suspected pain amplification syndrome	Improved pain Treatment de-escalation	No	No baseline	N/A
4	18y Female	Lower limb 6 Pelvic 1	NSAID Opioid	Pain	18 months of Treatment with Zoledronate Transferred to adult services—no further data	Improved pain Treatment de-escalation	N/A	No baseline	No follow-up scan
5	12y Female	Lower limb 3	NSAID Steroid	Pain	18 months of treatment with Zoledronate Treatment complete	Improved pain Treatment de-escalation	N/A	LS BMAD +1 TBLH BMD +0.5	No follow-up scan
6	13y Female	Severe spinal 2 Mild spinal 1 Upper limb 2 Lower limb 4 Pelvic 2	NSAID	Spinal lesions	30 months of treatment with Zoledronate Treatment complete	Treatment de-escalation	No	LS BMAD + 1.7 TBLH BMD +0.9	LS BMAD + 0.1 per year TBLH BMD +0.7 per year
7	11y Male	Lower limb 4 Pelvic 1	Missing data	Pain	30 months of treatment Pamidronate—every 3 months then every 4 months Treatment complete	Improved pain	N/A	No baseline	No follow-up scan
8	10y Female	Lower limb 5 Clavicle 1	NSAID Steroid	Pain	18 months of treatment with Zoledronate Treatment complete	Improved pain Treatment de-escalation	N/A	LS BMAD +2.1 TBLH BMD +1.6	No follow-up scan
9	12y Male	Mild spinal 3 Lower limb 5 Pelvic 1	NSAID Steroid DMARD	Spinal lesions Pain	30 months of treatment Pamidronate for 12 months then zoledronate for 18 months Treatment stopped due to BMD increase	Improved pain Treatment de-escalation	None	No baseline scan	Not known * [When checked at Year 3:* *LS BMAD +3.9* *TBLH BMD +2.5]*
10	14y Male	Severe spinal 2 Moderate Spinal 2 Mild spinal 8 Lower limb 5	NSAID	Spinal lesions Pain	18 months of treatment with Zoledronate Treatment complete	Improved pain	None	LS BMAD +1.6 TBLH BMD +0.3	LS BMAD −0.3 per year TBLH BMD −0.1 per year
11	9y Female	Plana spinal 1 Severe spinal 2 Upper limb 1 Lower limb 4	NSAID	Spinal lesions	Ongoing treatment Zoledronate Doses halved due to escalating BMD	Not known (data missing)	No	LS BMAD +2.6 TBLH BMD +0.4	Not known (in first year of treatment)
12	7y Female	Clavicle 1	NSAID Steroid DMARD	Pain	27 months of treatment Zoledronate dose interval increased for escalating BMD Treatment stopped due to escalating BMD	Improved Pain Treatment de-escalation	N/A	LS BMAD + 0.8 TBLH BMD +0.3	LS BMAD + 0.3 per year TBLH BMD +0.3 per year
13	16y Male	Plana spinal 1 Moderate spinal 2 Upper limb 1 Lower limb 1 Pelvic 1	NSAID Opioid	Spinal lesions Pain	12 months of treatment with Zoledronate Treatment complete Suspected pain amplification syndrome	Improved pain Treatment de-escalation	Yes (except plana lesion)	LS BMAD + 1.4 TBLH BMD +0.6	N/A
14	11y Male	Severe spinal 3	NSAID Opioid	Spinal lesions Pain	18 months of treatment (ongoing) Zoledronate Doses halved for escalating BMD	Improved pain Treatment de-escalation	Yes	LS BMAD + 1.3 TBLH BMD +1.1	LS BMAD + 0.65 per year TBLH BMD +0.45 per year
15	16y Male	Upper limb 1 Pelvic 1	NSAID Steroid	Pain	18 months of treatment Pamidronate for 6 months then zoledronate for 12 months Treatment complete	Improved pain Treatment de-escalation	N/A	No baseline scan	No follow-up scan
16	14y Female	Severe spinal 2 Mild spinal 1 Clavicle 1	NSAID Steroid	Spinal lesions Pain	6 months of treatment (ongoing) Zoledronate Transferred to another region	Improved pain Treatment de-escalation	N/A	No follow-up scan	No follow-up scan
17	12y Male	Plana spinal 3	NSAID	Spinal lesions Pain	30 months of treatment with Zoledronate Interval increased for escalating BMD Treatment complete	Improved pain Treatment de-escalation	No	LS BMAD + 1.5 TBLH BMD +1.1	No follow-up scans
18	10y Female	Lower limb 2 Jaw 1	NSAID Opioid	Pain	Zoledronate—due to receive first dose	N/A (treatment not yet received)	N/A	N/A	Not yet due
19	12y Female	Lower limb 4 Pelvic 1 Clavicle 1	NSAID	Pain	36 months of treatment Pamidronate for 12 months then zoledronate for 24 months Treatment complete	Improved pain Treatment de-escalation	N/A	LS BMAD + 0.8 TBLH BMD +0.4	No follow-up scan
20	14y Female	Lower limb 3 Pelvic 2	NSAID	Pain	12 months of treatment with Pamidronate Treatment discontinued—no improvement in pain Suspected pain amplification syndrome	None	N/A	No baseline scan	No follow-up scan

BP = bisphosphonate, BMD = bone mineral density, BMAD = bone mineral apparent density, DMARD = disease-modifying anti-rheumatic drug, NSAID = non-steroidal anti-inflammatory, TBLH = total body less head, LS = lumbar spine.

## Data Availability

The original data is publicly unavailable due to privacy or ethical restrictions. Summary data is included within the article.

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
