# Peer review of "Clinical, Bone Mineral Density and Spinal Remodelling Responses to Zoledronate Treatment in Chronic Recurrent Multifocal Osteomyelitis"

_diagnostics, 2025, doi:10.3390/diagnostics15182320_

Round 1

Reviewer 1 Report

Comments and Suggestions for Authors

Dear authors,

First of all, I congratulate you on your study. The effectiveness of zoledronate in the treatment of CRMO is impressively presented. The nearly 10-year follow-up is extremely valuable. I believe this is an innovative treatment and will be groundbreaking in this field. I read your manuscript with pleasure and liked it very much. However, I would like to ask you to provide two places.

1- Provide more figures showing the effect of zoledronate treatment on bones

2-Your tables are too complicated. Make them more understandable.

Author Response

Comment 1- Provide more figures showing the effect of zoledronate treatment on bones

Thank you for this valuable feedback, I have now included a third example as part of figure 1. 

Comment 2 - 2-Your tables are too complicated. Make them more understandable.

Again, thank you for the feedback, I have now tidied the table formatting and cut-down on the information to make it more understandable. 

Reviewer 2 Report

Comments and Suggestions for Authors

Chronic Recurrent Multifocal Osteomyelitis (CRMO) is a rare auto-inflammatory disease affecting the growing skeleton and leading to significant impairment in quality of life for children and adolescents. With increasing recognition, CRMO appears to be more common than previously thought, potentially as frequent as bone infections.

Nonsteroidal anti-inflammatory drugs (NSAIDs) are usually the first-line treatment for patients without vertebral column involvement. However, NSAIDs alone are insufficient for some CRMO patients, and additional therapies are required.

Bisphosphonates, which inhibit bone resorption, have pain modifying effects and anti-inflammatory properties. Evidence from small case series and retrospective studies suggests that bisphosphonates are effective in NSAIDs-refractory CRMO, often causing a rapid reduction in pain and even complete remission of symptoms.

In this manuscript, the authors present a retrospective study of pediatric patients with CRMO who received bisphosphonates (pamidronate and/or zoledronate) between 2008 and 2023 at a single tertiary center, Birmingham Women's and Children's Hospital. The study assessed treatment response in terms of pain, spinal remodeling and bone mineral density (BMD).

All patients in this cohort had refractory pain and tolerant of NSAIDs treatment. The analysis showed that 77% (n=14/18) reported improvement in pain, and 44% (n=7/16) had their NSAIDs use reduced or discontinued. The two non-responders to bisphosphonate were later found to have pain amplification syndrome. Three patients required treatment escalation during or after bisphosphonate therapy due to suspicion of a monogenic mimic of CRMO or other causes. Spinal remodeling was observed in 38% of patients with spinal lesions, but there was no radiological evidence of improvement in any vertebra plana lesion.

The greatest increase in BMD occurred during the first year of treatment, particularly in spine. Six patients demonstrated rapidly increasing BMD of five of them reached supraphysiological levels. This raises safety concerns when bisphosphonates are used in patients with a normal BMD. The authors therefore recommend using lower doses than those for osteoporotic conditions and tailoring therapy base on baseline and serial bone densitometry, along with clinical assessment of pain and lesion progression.

This is an excellent retrospective study and the first large-cohort report of CRMO patients treated predominantly with zoledronate, focusing on its effects on BMD and spinal remodeling. As a retrospective study, it has some limitations, including the absence of a validated pain score before and after treatment, variation in imaging modalities for vertebral remodeling analysis, and incomplete baseline and annual follow up BMD scans.

Overall, the manuscript is well-structure and clearly written. The classification and analysis of patients are detailed and reliable. The only limitation is the relatively small number of bisphosphonate-treat cases (n=20).

Comment:

I am not sure if the messy layout of Table 1 is due to typesetting or another reason, but it should be adjusted for clarity.

Author Response

Comment 1: I am not sure if the messy layout of Table 1 is due to typesetting or another reason, but it should be adjusted for clarity.

Response : Thank you for your detailed feedback and review. You are correct that the typesetting had caused a display issue with my table. I have now updated it to make it more presentable and clear.